# Entropy and interfacial energy driven self-healable polymers

Chris C. Hornat[1] & Marek W. Urban [1✉]

Although significant advances have been achieved in dynamic reversible covalent and non-covalent bonding chemistries for self-healing polymers, an ultimate goal is to create high strength and stiffness commodity materials capable of repair without intervention under ambient conditions. Here we report the development of mechanically robust thermoplastic polyurethane fibers and films capable of autonomous self-healing under ambient conditions. Two mechanisms of self-healing are identified: viscoelastic shape memory (VESM) driven by conformational entropic energy stored during mechanical damage, and surface energy/tension that drives the reduction of newly generated surface areas created upon damage by shallowing and widening wounds until healed. The type of self-healing mechanism is molecular weight dependent. To the best of our knowledge these materials represent the strongest ($S_f = 21$ mN/tex, or $\sigma_f \approx 22$ MPa) and stiffest ($J = 300$ mN/tex, or $E \approx 320$ MPa) self-healing polymers able to repair under typical ambient conditions without intervention. Since two autonomous self-healing mechanisms result from viscoelastic behavior not specific to a particular polymer chemistry, they may serve as general approaches to design of other self-repairing commodity polymers.

---

[1] Department of Materials Science and Engineering, Center for Optical Materials Science and Engineering Technologies (COMSET), Clemson University, Clemson, SC 29634, USA. ✉email: mareku@clemson.edu

Continuous interests in developing self-healable polymers are driven by the desire to extend life spans of existing functional materials. Environmental impacts stemming from the slow degradation of commodity polymers have accentuated the need for sustainable commodity materials[1], including polymers that self-repair after damage. Several reviews summarized recent advances[2–6]. Over the last two decades self-healing studies predominantly concentrated on incorporating specific physical and chemical repairing components, but recently van der Waals (vdW) interactions[7,8] and microphase separation[9,10] offered new directions for self-healable commodity thermoplastics without elaborate chemical modifications. Although molecular processes that govern self-repair in these materials are not easily measured, spectroscopic and mechanical analyses combined with molecular dynamics (MD) simulations showed that self-healing can be facilitated by the design of copolymer topologies and/or processing conditions. An ultimate challenge is to develop mechanically robust and affordable materials with high strength and stiffness that can autonomously and repetitively self-repair without external intervention.

In the context of these developments, other studies have also shown that polymers exhibit viscoelastic length transitions (VLTs) near the glass transition ($T_g$)[11]. These unique shape memory transitions are macroscopically manifested by directional extension and subsequent retraction in thermoplastic and thermosetting polymers during dynamic mechanical analysis (DMA). Viscous components of the network at the onset of the $T_g$ are responsible for polymer extension, whereas retraction results from stored conformational entropy as a result of chemical/physical crosslinks and/or chain entanglements. This behavior is quantified in terms of both strain ($\varepsilon_{max}$) and stress ($\sigma_{SF\ at\ \varepsilon_{max}}$) aspects through stored ($\Delta S_S$) and released ($\Delta S_R$) entropic energy densities, enabling relative measurements of SMEs.

While shape memory assisted self-healing (SMASH) methods utilizing reversible plasticity shape memory (RPSM) have proven effective in facilitating wound closure[10,12–17], application of heat or other stimulus is required to initiate repair. Therefore, damage must be detected and appropriately treated in a timely manner to prevent propagation and ultimate failure. These studies test the hypothesis that by taking advantage of the continuous viscoelastic nature of the glass transition, sufficient molecular mobility for autonomous self-healing under ambient conditions can be achieved while maintaining high strength and stiffness in precisely designed commodity polymers through efficient storage and recovery of conformational entropic energy using viscoelastic shape memory (VESM). Furthermore, excess surface energy resulting from damage can drive self-healing without intervention in low molecular weight polymers lacking sufficient junction density ($\nu_j$) for entropic memory of the undamaged geometry. An ultimate aim is to elucidate how molecular events during the damage-repair cycle contribute to macroscopic wound healing without external intervention in polymeric fibers and films under ambient conditions. These advancements are critical to many applications ranging from self-healable fabrics to optical fibers or self-healable coatings and films. Due to versatile applications and tailorable chemical compositions, polyurethanes are ideal to test the hypotheses that conformational entropy and surface energy may facilitate self-healing in polymers.

## Results

**Macroscopic damage-repair cycle.** Figure 1a, b illustrate optical images of thermoplastic polyurethane (TPU) fibers with $M_W \approx$ 72 kDa and 45 kDa, respectively, immediately after mechanical damage (A1 and B1), 40 min (A2 and B2), and 24 h (A3 and B3) after repair at 25 °C and ~50% relative humidity (RH). As seen, mechanical damage of 72 kDa fibers visually vanishes within

40 min (A2). Note the kinetics of self-healing may be affected by RH; for example, ~1.1 wt% $H_2O$ will lower the $T_g$ by ~7 °C, thus increasing chain mobility. However, lower molecular weight fibers (≤45 kDa) do not self-heal within this time frame, which is counterintuitive as one would anticipate that lower molecular weight with lower $T_g$ should be more conducive to flow and self-repair[3]. To evaluate self-healing efficiency, stress-strain measurements (Fig. 1c) before damage (a) and after self-repair (b) were conducted. While mechanical properties of TPU fibers with $M_W \approx 72$ kDa (C1) are completely recovered, lower molecular weight ($M_W \leq 45$ kDa) TPU fibers (C2) exhibit a significant decrease in tenacity ($\sigma_{failure}$) and failure strain ($\varepsilon_{failure}$) relative to their undamaged state. The corresponding mechanical properties of undamaged and repaired TPU fibers (Supplementary Table 1) as well as experimental details regarding synthesis, characterization, and analysis are provided in the Supplementary Materials.

**Shape memory VLTs and self-healing.** Shape memory viscoelastic length transitions (VLTs)[11] detected by measuring strain ($\varepsilon$) as a function of temperature during DMA are illustrated in Fig. 1d for 22, 32, 45, 72, and 180 kDa molecular weight ($M_W$) TPU. Building from thermodynamics of molecular network and rubber elasticity theory[18,19], the maximum strain ($\varepsilon_{max}$), stress ($\sigma_{SF}$ at $\varepsilon_{max}$), and entropic energy density stored ($\Delta S_S$) and released ($\Delta S_R$) during VLTs were determined. The decrease in conformational entropy (per volume) upon uniaxial extension is expressed as

$$\Delta S = -\frac{\nu_j R}{2}\left[\alpha^2 + \frac{2}{\alpha} - 3\right] \tag{1}$$

where: $\alpha$ is the extension ratio ($\alpha = L/L_o$), $\nu_j$ is the junction density ($\nu_j = \rho/M_j$), and $R$ is the gas constant. The corresponding entropic restoring force ($F_R$) generated can be expressed as

$$\frac{F_R}{A} = \sigma_R = \nu_j RT\left[\alpha - \frac{1}{\alpha^2}\right] \tag{2}$$

where: $T$ is temperature, $A$ is cross-sectional area, and $\sigma_R$ is the analogous retractive stress. As molecular weight decreases, $\varepsilon_{max}$ and $\varepsilon_{min}$ increase, while recovery percentage and released conformational entropic energy density ($\Delta S_R$) decrease due to reduced efficiency in storing and releasing conformational entropic energy (Supplementary Table 2). Since molecular entanglements act as junction-points for shape memory induced recovery[20] in amorphous thermoplastics, shape recoverability and energy storage upon deformation or damage are limited by increased chain slippage and flow as molecular weight (and $\nu_j$) decreases. These results are consistent with the self-healing behavior of TPU films of the same molecular weights (Supplementary Fig. 3). It is also useful to compare the VLT $\varepsilon_{max}$, $\sigma_{SF}$ at $\varepsilon_{max}$, and $\Delta S_S$ values to other polymers. As illustrated by plotting these values on the polymer shape memory prediction plane[11] shown in Fig. 1e, 180 kDa and 72 kDa TPU possess balanced $\sigma$ and $\varepsilon$ storage capabilities as well as high $\Delta S_S$ compared to other polymers owing to their $\nu_j$ (456.3 and 321.1 mol/m³) and tan $\delta_{max}$ (1.75 and 1.79) values. In contrast, 45, 32, 22 kDa TPU do not fall on the plane due to poor entropic recovery.

Differing from conventional shape memory and RPSM where deformation is fixed and sustained indefinitely until application of a stimulus[14,21,22], the distinctive macroscopic shape recovery behavior of 72 kDa TPU is illustrated in Fig. 1f by the isothermal viscoelastic shape memory cycle (VE-SMC) (21 °C, RH = ~50% RH), which shows that spontaneous gradual recovery occurs over approximately 2 h under these conditions. Importantly, wound closure under the same conditions requires 2 h as well (Supplementary Fig. 4B). While heat can be applied

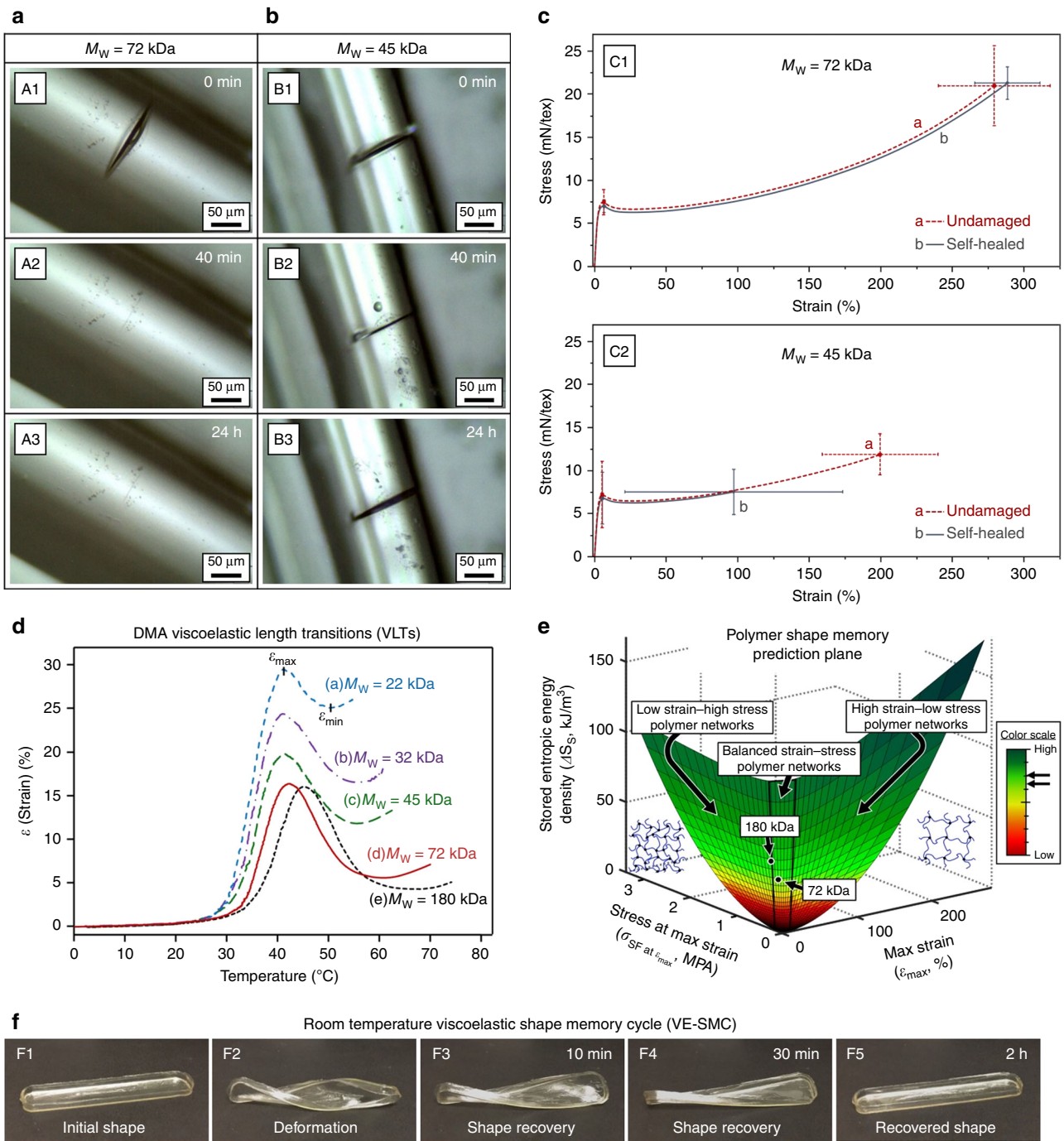

**Fig. 1 Self-healing and shape memory properties of TPU. a, b** Optical images of damaged TPU fibers: $M_W \approx 72$ kDa (A1-A3), and $M_W \approx 45$ kDa (B1-B3). Fibers were allowed to heal at 25 °C and ~50% RH. **c** Tensile stress-strain curves for $M_W \approx 72$ (C1) and 45 kDa (C2) TPU fibers before damage (**a**) and following repair for 7 days (**b**). **d** DMA strain ($\varepsilon$) curves showing the viscoelastic length transition (VLT) measured as a function of TPU molecular weight: $M_W \approx 22$ kDa (**a**), $M_W \approx 32$ kDa (**b**), $M_W \approx 45$ kDa (**c**), $M_W \approx 72$ kDa (**d**), and $M_W \approx 180$ kDa (**e**). **e** 180 and 72 kDa TPU VLTs maximum strain ($\varepsilon_{max}$), stress at maximum strain ($\sigma_{SF}$ at $\varepsilon_{max}$), and stored entropic energy densities ($\Delta S_S$) on the DMA shape memory prediction plane. **f** Isothermal viscoelastic shape memory cycle (VE-SMC) of 72 kDa TPU film (deformation and recovery both at 21 °C, ~50% RH).

to initiate faster recovery in $T_g$-based shape memory polymers (SMPs), viscoelastic recovery from deformation under ambient conditions is due to the close proximity of the $T_g$ (Supplementary Fig. 5A). This not only allows the material to mitigate cracking/fracture[13] as a result of viscoelastic toughening[23] and efficiently store conformational entropy when deformed/damaged under ambient conditions, but also gradually recover autonomously without sacrificing high strength and stiffness.

**Molecular events during damage-repair cycle.** To examine the molecular events responsible for wound closure and mending and how they translate to self-healing during the damage-repair cycle, internal reflection infrared imaging (IRIRI) was employed. Analysis of urethane amide I, II, and III bands at 1695, 1533, and 1241 cm$^{-1}$, respectively, as a function of distance from the wound (Supplementary Fig. 6A, C and D) show that these bands decrease following damage, but return to their

original undamaged intensities following self-repair (Supplementary Fig. 7A', B'). Thus, no detectable covalent bond cleavage or reformation occur upon damage or repair, but reversible conformational changes result from chain deformations. Furthermore, H-bonding dissociation inside the wound is manifested by the increase of the 1724 cm$^{-1}$ band due to non H-bonding C=O groups and decrease of the 1683 cm$^{-1}$ band due to H-bonding C=O (Supplementary Fig. 6B). 2D-FTIR correlation spectroscopy[10,24,25] was utilized to resolve these intensity changes in underlying bands from H-bonding and free urethane groups by using the damage-repair cycle as the perturbation stimulus. Synchronous 2D-FTIR spectra (Fig. 2, A1-A3 and B1-B3) show that upon both damage (A1-A3) and repair (B1-B3) H-bonding urethane amide I, II, and III bands (near 1682 cm$^{-1}$, 1540 cm$^{-1}$, and 1244 cm$^{-1}$, respectively) display positive cross-peaks with respect to each other, but negative cross-peaks with free urethane amide I, II, and III bands (1719 cm$^{-1}$, 1505 cm$^{-1}$, and 1224 cm$^{-1}$). These results indicate that H-bonding and non H-bonding bands exhibit opposite responses throughout the damage-repair cycle. Asynchronous spectra (Fig. 2, A4-A6 and B4-B6) show that upon damage (A4-A6) bands associated with free urethane increase relative to H-bonding, demonstrated by positive cross-peaks (1683 cm$^{-1}$, 1716 cm$^{-1}$), (1537 cm$^{-1}$, 1508 cm$^{-1}$), and (1244 cm$^{-1}$, 1225 cm$^{-1}$). However, during repair (B4-B6) free urethane bands decrease relative to H-bonding, signified by negative cross-peaks (1683 cm$^{-1}$, 1716 cm$^{-1}$), (1537 cm$^{-1}$, 1508 cm$^{-1}$), and (1244 cm$^{-1}$, 1225 cm$^{-1}$).

Given the contribution of macromolecular entanglements to mechanical properties of linear polymers and the lack of covalent bond formation, interdiffusion also plays a role in TPU self-repair, as demonstrated by complete recovery of failure stress and strain (Fig. 1, C1). Analysis of relaxation times ($1/\omega_c$) in rheological measurements from the storage modulus (E')/loss modulus (E") crossover frequency ($\omega_c$) shows that the efficacy of self-healing is greater when relaxation times are short due to greater chain mobility[26]. At 25 °C and ~50% RH, 72 kDa TPU exhibits self-repair (Supplementary Fig. 4A), and $\omega_c$ occurs at 0.27 Hz (Supplementary Fig. 8A), corresponding to a relaxation time ($1/\omega_c$) of 3.7 s. However, at 21 °C and <10% RH, TPU does not self-heal (Supplementary Fig. 4C), and does not exhibit the E'/E" crossover point within the 200 to 0.01 Hz frequency range (Supplementary Fig. 8B).

Progression of wound closure was followed using 3D-laser microscope surface profiling (Fig. 3a). As shown, the wound edges are pushed up and out from the center upon damage, but return to their initial positions during repair due to shape-recovery, reestablishing contact at the interface. These visual observations combined with the preceding spectroscopic and mechanical analyses demonstrate that for polymers with sufficient molecular weight and junction density ($v_j$) for efficient entropic-recovery, self-repair consists of two sequential/overlapping events schematically depicted in Fig. 3b. Upon damage (B2, and B2') H-bonding decreases in the wound and deformation causes chain conformational changes, manifested by conformational entropy decreases near damaged areas (Fig. 1 d-f, Supplementary Fig. 6A, C and D), generating the restoring force responsible for shape recovery[4,10,11,14,18]. In contrast to previous methods requiring a stimulus to trigger repair[10,12–17], the entropic force then brings the two cut surfaces back into contact without intervention under ambient conditions, physically closing the wound (Fig. 3 B3-B4, B3'-B4'). Once the two surfaces are in contact, H-bonding reformation (Fig. 2b) and chain diffusion[27,28] (Supplementary Fig. 8) facilitate re-bonding of the interface to restore mechanical integrity (Fig. 1 C1, Fig. 3 B5). Mechanical analysis showed that this TPU is the strongest ($S_f = 21$ mN/tex, or

$\sigma_f \approx 22$ MPa) and stiffest ($J = 300$ mN/tex, or $E \approx 320$ MPa) self-healing polymer capable of complete and autonomous repair under ambient conditions.

As shown in Fig. 1b, lower molecular weight (lower $v_j$) polymers (in this case, below ~45 kDa M$_W$) do not self-heal within the anticipated timeframe of a few hours due to poor entropic recovery. However, extended time (days) may lead to self-healing by a different mechanism. Optical images in Fig. 3c, A-C for 22, 32, and 45 kDa demonstrate that the lowest molecular weights repair faster (22 kDa → ~8 days), while 45 kDa is only partially healed even after 100 days. The side-view progression of self-healing as a function of time for a 22 kDa fiber is illustrated in Fig. 3d. Instead of closing, wounds widen and shallow over time until healed. Thus, the mode of self-repair is entirely different from the entropy-driven mechanism described in Fig. 3b. We hypothesize that excess surface energy[29–31] acts to reduce newly generated surface area created upon damage by inducing interfacial flow, leading to self-healing. The magnitude of the surface energy/tension driving force, or stress from surface tension ($\sigma_{ST}$), is a function of the curvature of the surface. At the bottom of a scratch, which has curvature in one direction, $\sigma_{ST}$ can be determined by $\sigma_{ST} = \gamma/R$ (where $\gamma$ is the surface tension coefficient of the material, and $R$ is the radius of curvature at the bottom of the scratch).

**Surface tension and self-healing.** To quantitatively examine dimensional changes as function of time during surface energy/tension driven repair, 3D-laser microscope surface profiling was utilized. Figure 4 a-d illustrates representative examples of laser microscope images of a scratch (32 kDa TPU fiber) over time (21 °C; ~50% RH) along with corresponding surface profiles overlaid in Fig. 4 e. The radius of curvature at the bottom of the scratch, corresponding driving force ($\sigma_{ST}$), scratch depth, and cross-sectional area of the wound (proportional to its volume) are plotted as functions of time in Fig. 4 f-i, respectively. $\sigma_{ST}$ values were determined by estimating $\gamma \approx 0.04$ N/m based on TPU chemical structure from molecular parachor (P$_S$)[32–34], where: $\gamma \approx (P_s/V_m)^4$ and $V_m$ is molar volume. Initially, the radius of curvature increases exponentially with time and $\sigma_{ST}$ changes inversely proportional to radius, while depth and cross-sectional area (volume) decrease exponentially. However, the kinetics of self-healing change after approximately 26-31 days, illustrated by the inflection point in Fig. 4 f (radius of curvature). Our hypothesis is this transition occurs when the curvature grows sufficiently large as to encompass the full width/depth of the wound (Fig. 4 j), whereas initially most of the side-walls of the scratch are relatively flat in comparison to the bottom of the wound. At this point, the healing rate considerably decreases, exemplified by the slower rates of decrease in depth and cross-sectional area (volume), and slower increase in the radius of curvature. A schematic illustration of the progression of geometrical changes during surface energy driven repair are shown in Fig. 4 k. The self-healing process is completed once sufficient volume has flowed to fill the wound. The healing rate is a function of molecular weight and increases as molecular weight decreases (22 kDa ~18 days, 32 kDa ~100 days, 45 kDa TPU is not fully repaired after 100 days) because viscosity ($\eta$) resists flow, and $\eta$ is proportional to chain length to the power of ~3.4 when above the critical entanglement length[35].

## Discussion

Self-repair of thermoplastic polyurethane fibers and films can occur via one of two different physical mechanisms depending on molecular weight. The approach of utilizing VESM in $T_g$-based SMPs to facilitate autonomous self-healing under ambient or

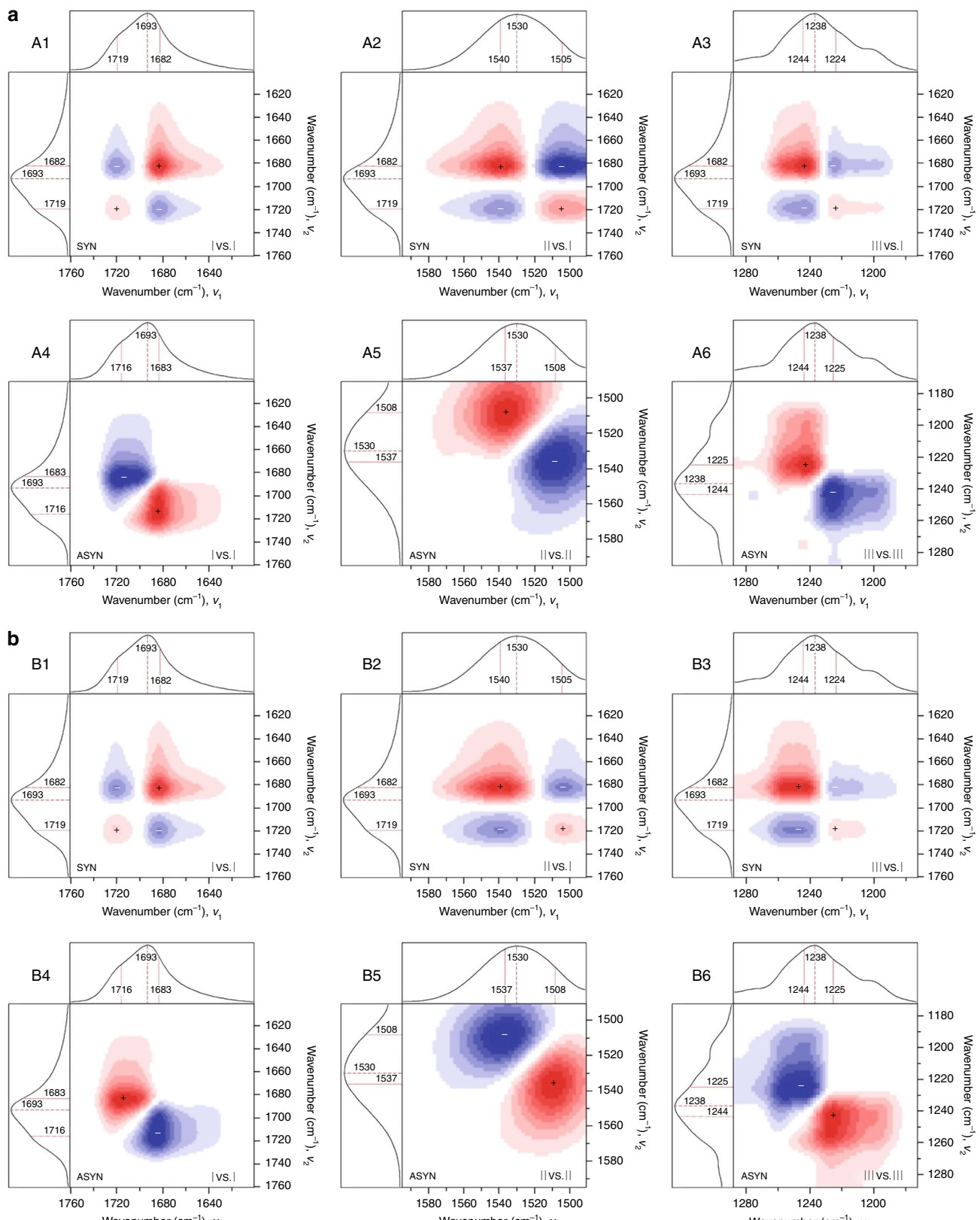

**Fig. 2 2D-FTIR correlation analysis. a** 2D-FTIR spectra of TPU going from the undamaged to damaged state, and **b** from the damaged to the healed state (μATR FT-IR). Synchronous (A1–A3 and B1–B3, using the average spectrum as the reference spectrum) and asynchronous (A3–A6 and B3–B6, using no reference spectrum). (A1 and B1) 1760–1600 cm$^{-1}$ (urethane amide I) vs. 1760–1600 cm$^{-1}$ (urethane amide I); (A2 and B2) 1595–1490 cm$^{-1}$ (II) vs. 1760–1600 cm$^{-1}$ (I); (A3 and B3) 1289–1173 cm$^{-1}$ (III) vs. 1760–1600 cm$^{-1}$ (I). (A4 and B4) 1760–1600 cm$^{-1}$ (I); (A5 and B5) 1595–1490 cm$^{-1}$ (II); (A6 and B6) 1289–1173 cm$^{-1}$ (III). (Red = positive, blue = negative).

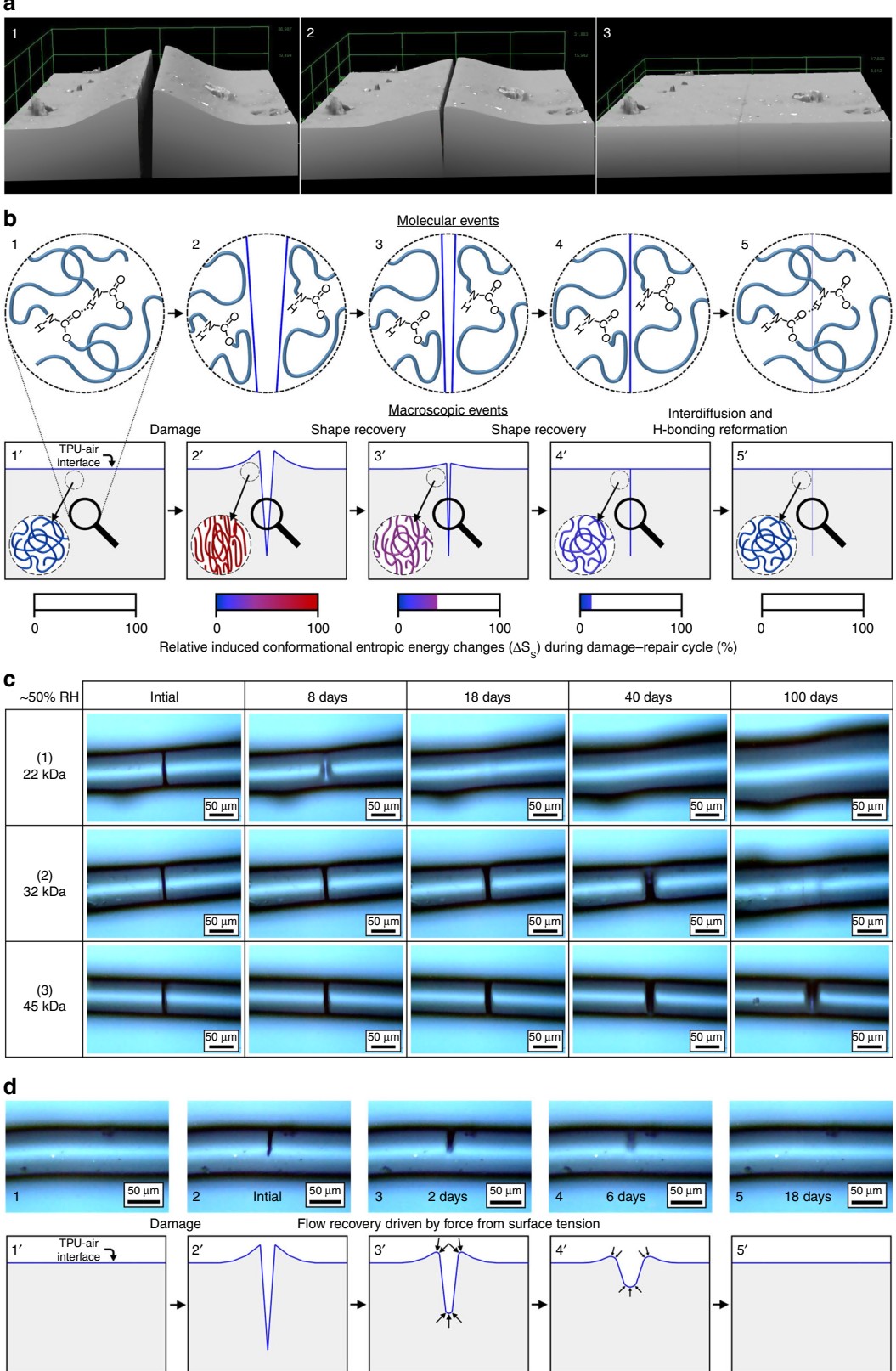

**Fig. 3 Proposed self-healing mechanisms. a** 3D-laser microscope surface profiles of 72 kDa TPU during wound closure. **b** Pictorial representation of molecular level (1–5) and macroscopic (1′–5′) events during entropy driven self-repair, illustrating chain conformational changes around the damage, breakage and formation of hydrogen bonds, and disentanglement and re-entanglement of chains during the process. **c** Optical images of surface tension driven self-healing in low molecular weight TPU: (1) $M_W \approx 22$ kDa; (2) $M_W \approx 32$ kDa; (3) $M_W \approx 45$ kDa (~21 °C, ~50% RH). **d** (1–5) Side-view optical images of surface tension driven repair in 22 kDa TPU fibers (~21 °C, ~50% RH); (1′–5′) schematic illustration of surface energy driven self-healing in lower molecular weight TPU.

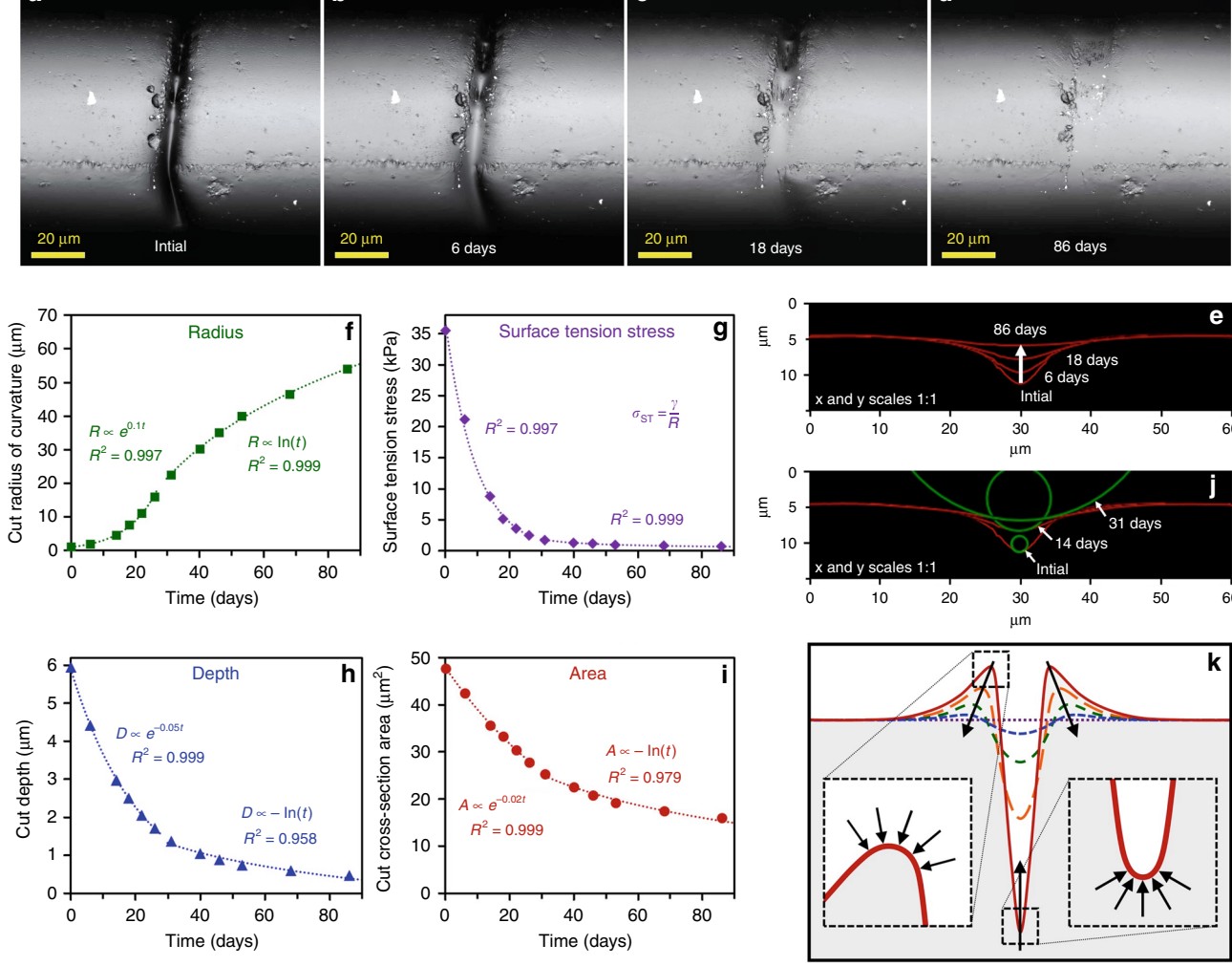

**Fig. 4** Laser microscope images of 32 kDa TPU fiber during surface tension/energy driven self-repair over time: **a** initial, **b** 6 days, **c** 18 days, and **d** 86 days after. Note that the "initial" images correspond to 9 days after damage, as prior to that point the scratch was too deep and narrow for the instrument to make quantitative measurements. **e** Overlay of corresponding scratch profiles from images A-D (laser microscope surface profiles). **f** Radius of curvature of the wound bottom, **g** the corresponding stress from surface tension ($\sigma_{ST}$) (TPU $\gamma \approx 0.04$ N/m), **h** cut depth, **i** and displaced cross-sectional area inside the scratch (proportional to volume) plotted as a function of healing time for 32 kDa TPU fibers (21 °C ~50% RH). **j** Overlay of the scratch profiles illustrating the change in radius of curvature at the bottom of the wound over time. **k** Schematic overlay of surface profile changes over time during surface tension driven self-repair.

other desirable conditions capitalizes on the abilities of entropic energy storage during damage and spontaneous gradual shape recovery during repair. These viscoelastic attributes allow adequate molecular mobility for wound closure and mending autonomously under ambient conditions while retaining high strength and stiffness. To the best of our knowledge, these materials exhibit the highest strength ($S_f = 21$ mN/tex, or $\sigma_f \approx 22$ MPa) and stiffness ($J = 300$ mN/tex, or $E \approx 320$ MPa) of self-healable polymers able repair at room temperature without requiring application of a stimulus. The striking shape recovery behavior can be effectively examined in a single DMA experiment by measuring VLT recovery percentage as well as stored ($\Delta S_S$) and released ($\Delta S_R$) conformational entropic energy density, which capture both strain ($\varepsilon_{max}$) and stress ($\sigma_{SF}$ at $\varepsilon_{max}$) aspects of relative shape memory capacity. These studies illustrate that efficacious facilitation of self-healing via shape memory methods requires efficient storage and release of conformational entropic energy during damage and repair. Consequently, this mechanism is ineffective for low molecular weight polymers lacking sufficiently entanglement/junction density to resist chain slippage/

flow upon damage/deformation. In contrast, lower molecular weight polymers (with low $v_j$) require extended times to self-heal through a separate process driven by interfacial/surface energy, which induces polymer flow to fill and repair the wound. Healing rate is therefore a function of viscosity, and self-repair will not occur in a reasonable period if viscosity is too high. One must weigh through the benefits of repair without intervention under ambient or use conditions against the drawback of potential viscoelastic creep that may unavoidably occur during autonomous self-healing of polymers, as there is an inherent conflict between the molecular dynamism needed for self-healing and polymer mechanical integrity/stability. Because these self-healing mechanisms are based on viscoelastic properties and not specific to a unique polymer chemistry, conformational entropy and surface energy can be utilized to create other commodity self-healing polymers, or combined with other self-healing chemistries to lead to new specialized materials. The simplicity and versatility of these approaches offer significant technological advantages for development of autonomous self-healing polymeric fibers, films, and coatings.

## Methods

**Materials.** Isophorone diisocyanate (IPDI) was purchased from Acros Organics. N, N-dimethylformamide (DMF) was acquired from Fisher Scientific. Poly-tetrahydrofuran (PTHF) ($M_n = 250$) and dibuyltin dilaurate (DBTDL) were purchased from Sigma-Aldrich Co. Traces of moisture were removed from PTHF drying at 65 °C in under vacuum for 24 h. All other chemicals were used as received.

**Fiber and Film Preparation.** Thermoplastic polyurethane was prepared by reacting IPDI with PTHF in DMF catalyzed by DBTDL with agitation from a stir bar at 600 rpm in a 20 mL vial immersed in an 80 °C oil bath under a $N_2$ atmosphere for 24 h. An excess of IPDI was used to compensate for reactions with absorbed water vapor. To produce 72 kDa TPU, the molar ratio of IPDI:PTHF was 1.06:1, while 1.04:1 was used for 45 kDa, 1.03:1 for 32 kDa, and 1.02:1 for 22 kDa. 180 kDa was also synthesized using a 1:06:1 ratio with brand new bottles of anhydrous DMF and IPDI, and the reaction was performed on a day with <10% RH in the lab, minimizing absorption of atmospheric moisture by DMF and PTHF during the setup of the reaction and weighing of components. The solutions were cast into PTFE molds and dried at 75 °C for 25 days. Fibers were manually pulled from the melt state using a glass pipette tip by heating to 200–170 °C (depending on MW, lower T for lower MW) for 15 s. Gel permeation chromatography (GPC) was performed to confirm molecular weight following fiber formation (see below), and thermogravimetric analysis (TGA) was performed to ensure all solvent was removed from both fibers and films prior to all other experiments. Fibers of 180 kDa TPU could not be produced, as the increased temperatures required for fiber formation of this molecular weight caused degradation and reduction of molecular weight to approximately 60 kDa.

**Fiber and film analysis.** Fibers were manually damaged by making scratches/partial cuts with a razor blade in the direction perpendicular to the fiber axis. Typical scratches/cuts were approximately ¼-½ of the fiber diameter in depth and no manual intervention was applied during self-healing process. Optical images were recorded using a Leica microscope (DM2500M) using WiRE version 3.4 software. 3D-laser microscope surface profiles, laser microscope images, 2D-cut profiles, and geometrical measurements were collected using an Olympus LEXT optical profiler/laser measuring microscope system. Images for self-healing time-lapse videos were taken using a Huvitz HRM-300 microscope and Panasis digital imaging software in 3D profile mode at a rate of 1 frame per minute. Relative humidity was measured using a Fisher Scientific Traceable Memory Hygrometer/Thermometer (11-661-18). For surface tension healing measurements in Fig. 3 c, d, immediately following damage fibers were subjected to high humidity (~90% RH) for 1–2 h (depending on molecular weight) prior to starting measurements in order to dissipate/relax any stored conformational entropic energy. This was done to prevent the wounds from initially closing partially via shape recovery, which occurs more prominently in higher molecular weights, and causes the cut geometries (in particular the initial ratios of the width to depth) to be inconsistent and different for each molecular weight. This approach resulted in consistent cut geometries at the start of the experiments, enabling comparison of healing rates as a function of molecular weight. The exposure time to high humidity for each molecular weight was chosen by determining the minimum time necessary for elimination of shape-memory type wound closure.

Microscopic attenuated total reflectance Fourier transform IR spectra (μATR FT-IR) were obtained using an Agilent Cary 680 FTIR single-beam spectrometer set at 4 cm$^{-1}$ resolution using a 2 mm diamond crystal, with constant contact pressure between crystal and specimen. All spectra were corrected for spectral distortions and optical effects using the Urban-Huang algorithm[36]. Spectra were analyzed using GRAMS AI software.

2D-FTIR correlation spectra[24] were calculated using 2DShige software (2Dshiege Shigeaki Morita, Kwansei-Gakuin University, 2004–2005). The average spectra was used as the reference for synchronous data, while no reference was used for asynchronous data. 2D-FTIR spectra in Fig. 3 are calculated from μATR FT-IR spectra. The damage-repair cycle was probed by first taking a spectrum of undamaged TPU, then making multiple cuts horizontally and vertically across the same spot prior to acquiring the spectra of "damaged" TPU. TPU was then given 7 days to repair prior to taking the measurement of "healed" TPU. All spectroscopic results shown were obtained using $M_W \approx 72$ kDa TPU.

Internal reflection IR images (IRIRI) were acquired using a Cary 600 series Stingray system equipped with internal reflection IR imaging providing ~1 μm spatial resolution[37]. The system comprises a Cary 680 spectrometer, a Cary 620 FTIR microscope, an image IR focal plane array (FPA) image detector, and a germanium (Ge) imaging crystal. The IR images were collected using under image sampling ratio 2, rapid-scan speed 5 Hz, number of images per step 64, 128 scans, and spectral resolution 8 cm$^{-1}$. Traces showing IR spectra inside and outside of damage were averaged over 24 spectra equidistant from the center of the wound. Spectra were interpolated using FFTx4 in GRAMS AI. IRIRI analysis was performed on the cross-section of cuts. Spectral subtraction in Supplementary Fig. 6 B was performed by first normalizing to 1695 cm$^{-1}$ band in order to see the ratio of the "free" (1722 cm$^{-1}$) and H-bonding (1695 cm$^{-1}$) C=O.

Dynamic mechanical analysis (DMA) was performed on a TA Instruments Q800 DMA in strain-controlled mode. The viscoelastic length transitions (VLTs) were measured under conditions specified in a previous study in order to obtain a relative measure of the shape memory effect (SME), and the shape memory prediction plane in Fig. 1e was generated as described previously and plotted in MATLAB[11]. (Note that VLTs are specific to DMA, and in particular to strain-controlled DMA). For temperature ramp experiments, rectangular film samples with dimensions of approximately 10 mm × 5 mm × 0.3 mm were run at a frequency of 10 Hz, with 10 μm amplitude and 125% force track ratio. Dynamic mechanical properties were measured from −20 °C up to 90 °C at 2 °C/min. For frequency sweep experiments, rectangular film samples with dimensions of 10 mm × 3 mm × 0.65 mm were run from 200 to 0.01 Hz with 7 points per decade, 10 μm amplitude and 125% force track ratio. Dynamic moduli and mechanical dampening were analyzed using TA Universal Analysis 2000.

Tensile testing was conducted using an Applied Test Systems Series 900 Universal Testing Machine with a 5 pound load cell according to ASTM D3822/D3822M on single polyurethane fibers. The average fiber diameter and linear density was 141 μm and 17.1 tex. The sample gauge length was 1 inch, and the cross-head speed was 2.4 inches/min. The temperature and relative humidity (RH) at the time of the experiment were 23.6 °C and 45% RH, respectively. Fiber diameters were determined by taking 5 micrographs along each fiber gauge length and measuring the diameter at each location using Image-Pro Plus 7.0 software and averaging. The linear densities were determined using a Thermo Orion Cahn C-33 microbalance to measure the fiber weights and a Mitutoyo 700-113-10 electronic digital caliper to measure the fiber lengths. Fibers were relaxed at 40 °C for 24 h prior to being damaged to remove any internal stresses. Fibers were given 7 days to heal at 25 °C and ~50% RH prior to tensile measurements. Non-damaged fibers were put through the same pre-test process. Results show in Fig. 1c and Supplementary Table 1 are averaged from 19 self-healed and 20 undamaged fibers that were tested. Error bars in Fig. 1c indicated standard deviations (SD).

Differential scanning calorimetry (DSC) curves were obtained using a TA Instruments Q1000 DSC, equipped with a liquid nitrogen cooling system. Samples were run from −100 °C to 160 °C at a heating rate of 15 °C/min. Thermogravimetric analysis (TGA) was conducted using a TA Instruments Q5000 TGA, with a heating rate of 15 °C/min from 25 to 600 °C in an N2 atmosphere. DSC and TGA data were analyzed using TA Universal Analysis software.

Gel permeation chromatography (GPC) measurements were conducted using a system with a Waters 717plus Autosampler, Waters 1525 Binary HPLC Pump, Waters 2414 Refractive Index Detector, Waters Styragel HR 5E 7.8 ×300 mm column, using HPCL grade Burdick & Jackson chloroform from Honeywell as the solvent and 1 mL/min flow rate. Polymer was given 24 h to dissolve, prepared at a concentration of 0.75 mg/mL of polymer in chloroform, prior to filtering with 0.2 μm Teflon membrane filters. To perform the measurements, 50 μL of solution was injected into the system. All molecular weight values were calibrated relative to narrow molecular weight polystyrene standards ranging from 1,000,000 to 436 Da in chloroform, and analysis was performed using Waters Breeze 2 Software.

## Data availability

All data needed to evaluate the conclusions in the paper are present in the paper or the supplementary materials.

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

## Acknowledgements

The authors thank the National Science Foundation under the award DMR 1744306 and J.E. Sirrine the Foundation for partial support of these studies. The authors also thank Prof. A. Ogale and J. Jin for assistance in performing tensile measurements, and K. Ivey for technical assistance in GPC measurements. Selected research reported in this publication was conducted using an Olympus LEXT OLS4100, housed in the Clemson Light Imaging Facility (CLIF). CLIF is supported, in part, by the Clemson University Division of Research, NIH EPIC COBRE Award #P20GM109094, and NIH SC Biocraft COBRE Award #5P20RR021949-03.

## Author contributions

The experiments were designed by M.W.U. and C.C.H. Experimental work was conducted by C.C.H. Data analysis was performed by M.W.U. and C.C.H. M.W.U. and C.C.H. wrote and edited the manuscript under supervision of M.W.U.

## Competing interests

The authors declare no competing interests.
