## [Peer Review File · Nature Communications]

Reviewers' comments:

Reviewer #1 (Remarks to the Author):

In this manuscript, Urban and coworkers developed a kind of mechanically robust thermoplastic polyurethane fibers and films capable of autonomous self-healing under ambient conditions. For high molecular weight polymers, self-healing is achieved through viscoelastic shape memory (VESM) driven by conformational entropic energy stored during mechanical damage; while for lower molecular weight polymers susceptible to flow, self-repair results from excess interfacial energy. This is a symmetrically study about how autonomous self-healing happens for thermal plastic polymers. This manuscript can be accepted for publication in Nature Communications after a minor revision. Here are some suggestions and comments.

1. Different molecular weight PUs were synthesized. Since the ratio of IPDI:PTHF was fixed at 1.06:1, how to control the molecular weight? I think it is difficult to control the molecular weight by tuning reaction time since the materials were annealed at 75 °C for 25 days. The condensation reaction will continue to go on until achieving equilibrium.

2. As can be seen from Figure S4, self-healing is achieved at 21 °C in about 50% RH, while no self-repair is observed at 21 °C in dry atmosphere (~0-10% RH). What's the role of humidity during the self-healing process?

3. I can fully understand the proposed different mechanisms for high and low molecular weight polymers. However for bulk materials, the cut-off samples were put together first, then after seven days self-healing was evaluated by tensile experiments. In this process, reformation of hydrogen bonds and interface diffusion should be responsible for the self-healing process, in this case why there is still different self-healing behavior happening for different molecular weight polymers as shown in Figure 1, C1 and C2.

4. The molar ratio of IPDI:PTHF was 1.06:1, which means that there is little free isocyanate left in the materials, does this left NCO group will contribute to the self-healing process? For example, NCO group will react with the water at 50% RH.

5. The stress-strain curves in Figure 1, C1 and C2 are very interesting, the stretching trend looks very similar, only for low molecular weight, short elongation at break was obtained. Does this happen suddenly, or this is indeed the case. How about other molecular weight polymers? If low molecular weight polymer can achieve the same modulus as high molecular weight polymer, perhaps different hydrogen bond aggregates are formed.

Reviewer #2 (Remarks to the Author):

Given the good academic quality of the senior author I have read the manuscript with attention.

However, much as I appreciate his work, I consider the submitted manuscript well below the level worthy to be published in Nature Communications for the following reasons:

- the self healing behaviour of PolyUrethanes as a result of hydrogen interactions (or other) has been demonstrated by many authors in the past
- the role of the entropic contribution in the driving force has been published by many authors, most notably Dr Urban himself
- the role of the surface tension as the driving force to make the crack disappear has been pointed out at least 10 year ago
- the reported observations of a crack closing focus on the less relevant features
- unlike recent measurements by other teams the current manuscript totally ignores the spatial dimensions of the strained region on either side of the scratch made.

Reviewer #3 (Remarks to the Author):

Linear thermoplastic polyurethane fibers and films are shown to undergo autonomous self-healing that is attributed to one or both of following mechanisms: (i) entropic recovery of viscoelastic memory that was created during mechanical damage and (ii) the spontaneous reduction in surface area that was created during mechanical damage. Material wound healing in the vicinity of the glass transition temperature is autonomous and fully recovers the material's remarkably high pristine strength and stiffness. More importantly, the proposed mechanisms appear to be general to polymers with similar viscoelastic behavior, forming a solid basis for materials design.

The collection of experimental data is original because it combines microscopy, tensile and dynamic mechanical testing, and 2D-FTIR spectroscopy to support conclusions. Data are interpreted on the basis of viscoelastic length transitions that have previously been reported (Hornat et al., *Advanced Materials*, 29, 2017, 1603334) and can quantify conformational entropic energy density from dynamic mechanical analysis temperature scans. Infrared imaging results provided insight into molecular events including H-bond dissociation inside the material wound; the 2D-FTIR spectra confirmed that damage generates free urethane absorption bands though healing consumes them. The analysis of surface repair by characterizing changes in surface geometry offers compelling evidence for surface tension driven repair. The data appear very consistent with the self-healing mechanisms offered, and, therefore the results will be valuable to the community by bringing new modes of thought to autonomous self-healing polymers.

The methods are well described and materials characterization appears straightforward and enough data is provided for others to reproduce experiments. The manuscript is well written and easy to follow.

Additional Comments

- One may expect viscoelastic creep to be inherent to polymers that autonomously self-heal, and creep could be a drawback for stress-bearing fibers.
- When interdiffusion was discussed, experiments were performed at ~50% relative humidity (RH) and ~10% RH with the implication that RH affects healing. A sentence to explain or provide a reference would be useful.
- Page 10, paragraph should begin "As shown"
- For low molecular weight systems, healing is driven by minimization of surface area. This implies that, upon contact, surfaces may self weld together, and interfacial adhesion between like surfaces may similarly occur.
- Some reference were incomplete, e.g. reference 11

Mitchell Anthamatten

Reviewer #1.

1) Different molecular weight PUs were synthesized. Since the ratio of IPDI:PTHF was fixed at 1.06:1, how to control the molecular weight? I think it is difficult to control the molecular weight by tuning reaction time since the materials were annealed at 75 oC for 25 days. The condensation reaction will continue to go on until achieving equilibrium.

The reviewer is correct that is difficult to control molecular weight and because this is steo-growth reaction sensitive to water and to control mol wt, we utilized different IPDI:PTHF ratios to achieve different molecular wts. Specifically, 1.06:1 ratio was used for 72 kDa TPU, 1.04:1 for 45 kDa, 1.03:1 for 32 kDa, and 1.02:1 for 22 kDa. 180 kDa was made also using 1:06:1 using a brand new bottles anhydrous DMF and IPDI, and reaction performed on a day with >10% RH, minimizing absorption of atmospheric moisture by DMF and PTHF during the setup of the reaction.”

The following was added to the Supp. Doc. page 2.

To produce 72 kDa TPU, the molar ratio of IPDI:PTHF was 1.06:1, while 1.04:1 was used for 45 kDa, 1.03:1 for 32 kDa, and 1.02:1 for 22 kDa. 180 kDa was also synthesized using a 1:06:1 ratio with brand new bottles of anhydrous DMF and IPDI, and the reaction was performed on a day with <10% RH, minimizing absorption of atmospheric moisture by DMF and PTHF during the set up of the reaction and weighing of components.

2) As can be seen from Figure S4, self-healing is achieved at 21 oC in about 50% RH, while no self-repair is observed at 21 □C in dry atmosphere (~0-10% RH). What’s the role of humidity during the self-healing process?

Response: This was addressed in the Supp. Doc. and reads as follows:

“The sensitivity of TPU self-healing on humidity is a result of the plasticization effect of water on polyurethanes ^{8,9}, as TPU absorbs ~1.1 wt% H₂O at ~50% RH (Supplementary Fig. 5B) which decreases T_g by ~7 □C (Supplementary Fig. 5A) and increases molecular mobility.” (Supplementary Material page 6, Optical Imaging section)

3) I can fully understand the proposed different mechanisms for high and low molecular weight polymers. However for bulk materials, the cut-off samples were put together first, then after seven days self-healing was evaluated by tensile experiments. In this process, reformation of hydrogen bonds and interface diffusion should be responsible for the self-healing process, in this case why there is still different self-healing behavior happening for different molecular weight polymers as shown in Figure 1, C1 and C2.

Response: Specimen were never manually put together following damage in any of the data presented. In all cases, samples were damaged, and conformational entropic energy or surface energy contributions acted to close or shallow the wound autonomously. In the case of conformational entropic energy facilitated repair (via

VESM), *after* the wound closed reformation of hydrogen bonds and intermolecular diffusion were found to be responsible for mending of the interface. The results in Figure 1, C1 and C2 reflect the abilities of conformational entropic energy to facilitate wound closure in 72 and 45 kDa TPU, not the abilities of hydrogen bond reformation or interdiffusion in these materials.

To make this clear, the following text was added to the Supp. Docs. p. 2.

Typical scratches/cuts were approximately $\frac{1}{4}$ - $\frac{1}{2}$ of the fiber diameter in depth and no manual intervention was applied during self-healing process.

4. The molar ratio of IPDI:PTHF was 1.06:1, which means that there is little free isocyanate left in the materials, does this left NCO group will contribute to the self-healing process? For example, NCO group will react with the water at 50% RH.

Response: Excess isocyanate was added to compensate for side reactions such as the reaction of isocyanate with any water impurities in the solvent (DMF) and monomers, as well as isocyanate self-reactions, etc. This is common practice in polyurethane chemistry to achieve the highest conversion possible. To the best of our detection ability using NMR and IR spectroscopy, no unreacted isocyanate groups remain following completion of the drying process (25 days). However unreacted isocyanate groups could be seen after film/fiber formation for less than 1 day. It is also well established that remaining unreacted isocyanate groups have strong affinity to react with water vapor which subsequently consumes a second isocyanate group to form polyurea linkages.

To clarify this point, the following text was added to the Supp. Docs. p. 2.

An excess of IPDI was used to compensate for the reactions with water vapor.

5. (a) The stress-strain curves in Figure 1, C1 and C2 are very interesting, the stretching trend looks very similar, only for low molecular weight, short elongation at break was obtained. Does this happen suddenly, or this is indeed the case. (b) How about other molecular weight polymers? (c) If low molecular weight polymer can achieve the same modulus as high molecular weight polymer, perhaps different hydrogen bond aggregates are formed.

Response: (a) It is unclear on what is meant by the comment/question “Does this happen suddenly, or this is indeed the case.” Yes, for undamaged fibers the lower molecular weight (45 kDa) has shorter elongation at break. This was fairly consistent, as indicated by the similarly sized error bars for failure strain of undamaged fibers of both molecular weights. A fairly large sample size was used (20 fiber samples each) so we are confident that this difference is likely real (we do not know if this is what is meant by “does this happen suddenly”). Molecular weight is known to have some effect on the tensile properties of thermoplastics in general, especially amorphous materials which are reliant on entanglements. Also, the T_g of the two molecular weights differ slightly (higher MW = slightly higher T_g) and due to the close proximity of T_g to ambient conditions, it is possible some temperature effects related to deformability may play some role as well (see reference 23).

For 45 kDa fibers which had been damaged and “self-healed”, much lower failure strain at break was observed. The exact extent varied from fiber to fiber, with some failing almost immediately after beginning elongation while others failed much closer to the values of undamaged fibers. This is evidenced by the large failure strain error bar for this group of samples in Figure 1 C2, curve b.

(b) These two molecular weights (72 kDa vs 45 kDa) were chosen specifically to highlight the boundary where the difference in healing behavior-type was observed, and a large sample size was prioritized to ensure the reliability of the results. The main purpose of the tensile experiments was to verify that the 72 kDa fibers had in fact mended, as visual wound closure does not guarantee mending has also occurred. The degradation of properties in damaged/“healed” 45 kDa fibers after 7 days of healing provides contrast by illustrating the effect on properties when closure and mending do not occur/complete. As noted in the Experimental Methods, fibers of 180 kDa TPU were attempted but could not be produced, as the increased temperatures required for fiber formation of this molecular weight caused degradation and reduction of molecular weight to approximately 60 kDa.

(c) As shown in Supplementary Table 1, the modulus of 45 kDa molecular weight is slightly lower than 72 kDa (~300 vs. 267 mN/tex).

=====
==

Reviewer #2.

Given the good academic quality of the senior author I have read the manuscript with attention.

I will take exception of the above statement and say that this reviewer is neither professional nor constructive. Nevertheless, here are the responses.

1) the self healing behavior of PolyUrethanes as a result of hydrogen interactions (or other) has been demonstrated by many authors in the past.

Response: What’s the point? This manuscript is not about H-bonding as the focal point. This is like saying because H-bonding is one not even critical elements (it could be covalent or any other type of bonding), all papers that utilize H-bonding should be disqualified? How about other types of bonding? The primary focal point of this paper is on the physical mechanisms of autonomous wound closure for different molecular weights, being driven by conformation entropic energy resulting in viscoelastic shape memory (VESM), or interfacial/surface energy resulting in material flow. Hydrogen bonding changes were discussed in part to provide the complete picture of the damage-repair process but is not meant to be the novelty. The use of unique spectroscopic tools (IRIRI, 2D-IR) made the detection of H-bonding changes of interest and importance.

2) the role of the entropic contribution in the driving force has been published by many authors, most notably Dr Urban himself.

Response: Our group has published a study on T_m -based shape memory driven by entropic contributions (Adv. Mat). However, we have never conducted studies in a systematic matter as this one which correlate shape memory with self-healing. The same applies to cited references in which several studies mentioned the use of shape memory in self-healing but without any depth. References 12-17 are all examples found in the literature. In this study, we demonstrated for the first time the use of entropic energy stored during damage evert to facilitate wound closure autonomously under ambient conditions through viscoelastic shape memory (VESM) phenomenon. This is one of the focal points of the manuscript, which was achieved by taking advantage of the continuous viscoelastic nature of the T_g , and also allowed for higher strength and stiffness than had been achieved in any self-healing polymer capable of complete autonomous self-healing under ambient conditions. Furfhtermore, this is the first study showing that viscoelastic length transitions (VLTs) from DMA measurements have been directly used to identify shape memory contributions to self-healing.

3) the role of the surface tension as the driving force to make the crack disappear has been pointed out at least 10 years ago.

Response: The reviewer is plain wrong. To the best of our knowledge, we are unaware of anything the reviewer may be speaking off specifically. Perhaps it could have been mentioned but the peer-review literature does not justify his/her assessments.

4) the reported observations of a crack closing focus on the less relevant features.

Response: It is difficult to identify what he/she has exactly in mind. Based on the previous comments, he/she does not seem to understand (intentionally or unintentionally based on the opening statement) or accept the novelty these studies show. In the case of wound closure (entropic driven repair), we established the ability of the material self-heal based on its physical properties to store and recover entropic energy leading to recovery of undeformed shape, and how that information relates to the efficiency of wound closure. In addition, this work identified conditions under which wound closure/self-healing/shape recovery occur. In future studies, we may consider exploring other aspects of wound closure, such as determining boundary conditions under which self-healing may occur and examine the distribution of stress stored in the material, etc., but, as pointed out on numerous occasions, that was not the objective of this study.

5) unlike recent measurements by other teams the current manuscript totally ignores the spatial dimensions of the strain region on either side of the scratch made.

Response: Again, it is not clear what the reviewer is referring to. We are unaware of studies that closely examine the “spatial dimensions of the strain region on either side of the scratch made.”

However, these studies showed the spatial measurements of the deformed regions on either side of the scratch, as shown in Figure 3A from 3D-laser microscope surface profiles for entropic driven recovery. For surface/interfacial energy driven

repair, we identified the complete dimensions/geometry of the wounds over time and measure the changes quantitatively, shown in Figure 4.

In summary, as other reviewers acknowledged, the objectives of these studies were to establish the role of entropic energy to facilitate wound closure autonomously under ambient conditions through viscoelastic shape memory (VESM), and identify the effect of molecular weight on self-healing. Thus, there is very little in any value in the comments of this reviewer.

=====
==

Reviewer #3.

1) One may expect viscoelastic creep to be inherent to polymers that autonomously self-heal, and creep could be a drawback for stress-bearing fibers.

Response: We agree, there may be some level of viscoelastic creep, which is likely an unavoidable drawback of any material able to self-heal autonomously under ambient or use conditions. There is an inherent conflict between the molecular dynamism needed for self-healing and total mechanical stability. It will likely be dependent on the specific use case/application whether having the ability to autonomously self-heal without intervention is significantly beneficial and the degree of creep is within an acceptable range, or if complete mechanical stability is needed in which case self-repair would require application of a stimulus such as heat.

To make this clear, the following text was added to the Main Doc, end of p. 12.

One must weight though the benefits of repair without intervention under ambient or use conditions against the drawback of potential viscoelastic creep that may unavoidably occur during autonomous self-healing of polymers, as there is an inherent conflict between the molecular dynamism needed for self-healing and polymer mechanical integrity.

2) When interdiffusion was discussed, experiments were performed at ~50% relativev humidity (RH) and ~10% RH with the implication that RH affects healing. A sentence to explain or provide a reference would be useful.

Response: This is explained in the Supplementary materials, as mentioned in Response 2 to Reviewer #1. "The sensitivity of TPU self-healing on humidity is a result of the plasticization effect of water on polyurethanes, as TPU absorbs ~1.1 wt% H₂O at ~50% RH (Supplementary Fig. 5B) which decreases T_g by ~7 °C (Supplementary Fig. 5A) and increases molecular mobility." (Supp. Docs. page 6, Optical Imaging section)

3) Page 10, paragraph should begin "As shown"

Response: Yes, the correction has been made on page 10.

4) For low molecular weight systems, healing is driven by minimization of surface

area. This implies that, upon contact, surfaces may self weld together, and interfacial adhesion between like surfaces may similarly occur.

Response: To some extent, this may be true, given enough time and especially if under external force, or at elevated temperature/humidity, some surface welding can occur in the lower molecular weight systems. However, previous research has suggested that polymer surfaces are more amenable to healing/welding shortly following fracture [references 1 and 2 below], as chain ends tend to segregate to fracture surfaces [2], which is favorable for healing/welding [2, 3], as tethered chain ends and free chains have higher mobility than undamaged chains [3].

[1] F. Maes, D. Montarnal, S. Cantournet, F. Tournilhac, L. Corté, L. Leibler, Activation and deactivation of self-healing in supramolecular rubbers, *Soft Matter* 8 (5) (2012) 1681-1687.

[2] R.P. Wool, Self-healing materials: a review, *Soft Matter* 4 (3) (2008) 400-418.

[3] Y. Yang, M.W. Urban, Self-healing polymeric materials, *Chemical Society Reviews* 42 (17) (2013) 7446-7467.

Since this is more of commentary than criticism, no changes were made.

5) Some references were incomplete, e.g. reference 11

Response: Corrections will be made where needed.

REVIEWERS' COMMENTS:

Reviewer #1 (Remarks to the Author):

The author have fully addressed my question, it can be accepted for publication.

Reviewer #3 (Remarks to the Author):

I have read the reviews and the authors' response, and I feel the authors have adequately responded to the comments and criticisms offered by the reviewers. However, I would like to submit additional comments that focus on the response, and one additional criticism / comment that may be important.

Reviewers #1 and #3 both pointed out that the role of humidity needs to be clarified. The authors responded that the role of humidity it was already addressed in the Supp. Document. Since two reviewers pointed this out, other readers will be thinking the same thing, and I suggest moving the explanation to the main text.

In response to Reviewer #2, item 2, the authors describe the correlation between shape-memory and self-healing. The work here reminds me a lot of the SMASH system that the authors refer to, however, present finding focuses on memory involving the continuous viscoelastic nature of the Tg. The Tg must be precisely tuned, and the timescale of stress relaxation in the vicinity of the Tg depends on both temperature and stress. The central idea here is that at a fixed temperature, enough stress accumulates near the damage, and this accelerates stress relaxation to help close the wound. This is connected to my concern (I'm reviewer #3) that mechanical creep will be inherent to the system. I.e. over long times (or over short times at elevated temperatures), any stress in the drawn fibers should relax. If stress relaxation involves polymer diffusion out of entanglements then the mechanical properties of the polymer may irreversibly change.

Reviewer #2 also pointed out that surface tension can cause the crack to disappear, and this has been known for some time. The authors replied that the reviewer is "plain wrong" . I'm also unaware of any specific example where surface tension is the only driving force for self healing, so the response seems ok. On the other hand, surface tension has certainly played a role in important self-healing examples including those with encapsulents that release liquids. The liquids interact and adhere to surfaces as part of the self-healing mechanism.

High stresses and elongational flows are known to cause bond breakage. Electron spin resonance studies have shed lots of light on this topic. Solvent swelling can even cause main chain scission. The new question to address is: would mechanical cleavage of polymer bonds during the damage process affect the conclusions draw regarding VESM as the mechanism for self healing?

Reviewer #1 (Remarks to the Author):

The author have fully addressed my question, it can be accepted for publication.

Reviewer #3 (Remarks to the Author):

1. I have read the reviews and the authors' response, and I feel the authors have adequately responded to the comments and criticisms offered by the reviewers. However, I would like to submit additional comments that focus on the response, and one additional criticism / comment that may be important.

No need for response – comments.

2. Reviewers #1 and #3 both pointed out that the role of humidity needs to be clarified. The authors responded that the role of humidity it was already addressed in the Supp. Document. Since two reviewers pointed this out, other readers will be thinking the same thing, and I suggest moving the explanation to the main text.

Response: In an effort to maintain the flow of the main document the only logical place to address was to add a footnote. The following footnote was added in the main document at the bottom of p.4.

* - Kinetics of self-healing may be affected by RH; for example, ~1.1 wt% H₂O will lower the T_g values by ~ 7oC, thus increasing chain mobility.

3. In response to Reviewer #2, item 2, the authors describe the correlation between shape-memory and self-healing. The work here reminds me a lot of the SMASH system that the authors refer to, however, present finding focuses on memory involving the continuous viscoelastic nature of the T_g. The T_g must be precisely tuned, and the timescale of stress relaxation in the vicinity of the T_g depends on both temperature and stress. The central idea here is that at a fixed temperature, enough stress accumulates near the damage, and this accelerates stress

relaxation to help close the wound. This is connected to my concern (I'm reviewer #3) that mechanical creep will be inherent to the system. I.e. over long times (or over short times at elevated temperatures), any stress in the drawn fibers should relax. If stress relaxation involves polymer diffusion out of entanglements then the mechanical properties of the polymer may irreversibly change.

No need for response – comments.

4. Reviewer #2 also pointed out that surface tension can cause the crack to disappear, and this has been known for some time. The authors replied that the reviewer is "plain wrong" . I'm also unaware of any specific example where surface tension is the only driving force for self healing, so the response seems ok. On the other hand, surface tension has certainly played a role in important self-healing examples including those with encapsulents that release liquids. The liquids interact and adhere to surfaces as part of the self-healing mechanism.

Response. The reviewer is correct, this study shows for the first time experimentally that surface tension may be critical in self-healing. However, the statement that '*surface tension has certainly played a role in important self-healing examples including those with encapsulents that release liquids*' is only partially correct. The important property is surface wetting of the liquid phase; thus, not the surface tension but the surface tension differential (between liquid and solid) is critical to self-healing.

5. High stresses and elongational flows are known to cause bond breakage. Electron spin resonance studies have shed lots of light on this topic. Solvent swelling can even cause main chain scission. The new question to address is: would mechanical cleavage of polymer bonds during the damage process affect the conclusions draw regarding VESM as the mechanism for self healing?

Response. Elongations and stresses are not sufficient to cause the chain cleavage in elastomers like polyurethanes. The reviewer is referring to ESR without saying what the ESR detects. ESR detects free radicals generated by the chain cleavage. What we observed in all systems including this one is the chain slippage which is manifested and discussed in FT-IR and 2D correlation spectroscopy sections. We use ESR spectroscopy all the time, but at the sensitivity levels of 1×10^{-7} moles (which is very high), no free radicals (that's what the reviewer #3 is referreing to when he/she talks about the chain cleavage) resulting from chain cleavage are detected. TPUs are elastomeric materials and before any bonds break chain slippage occurs. Our recent studies (Science, 2018, 362, 220-225) explicitly have shown that free radical do not contribute to self-healing even in much more brittle systems. I am a bit confused how to respond to this comment. Should we start as new ESR study because this is what reviewer thought as an afterthought? Please advise.